

# Representation of investigation results of microplastics on sandy beaches—accumulation rate and abundance in the entire study site

Hiroshi Asakura

Institute of Integrated Science and Technology, Nagasaki University, Nagasaki, Japan

## ABSTRACT

Long-term microplastics (MPs) environmental pollution trends cannot be understood only by investigating their presence on beaches. Without estimating MPs for the entire beach, comparisons between multiple beaches cannot be made. In this study, Nagasaki Prefecture was selected as the study site, we measured MPs accumulation rate to express the MPs pollution trend and weighted the measurement results to enable comparison of MPs content among multiple sandy beaches. The MPs accumulation rate in the study site was measured by periodic investigation at fixed spots. The average in the supratidal zone was $1.5 \pm 0.9$ mg-MPs/(m$^2$-sand· d) ($n = 15$). The weighting of the MPs content in hot spots and non-hot spots by their respective areas enabled us to obtain the representative value and the dispersion of the MPs content in the entire study site. The MPs contents in the three beaches were $298 \pm 144$, $1{,}115 \pm 518$, and $4{,}084 \pm 2{,}243$ mg-MPs/(m$^2$-sand), respectively. Using these values, it is possible to compare the MPs contents of multiple beaches.

## INTRODUCTION

Global plastic waste pollution is occurring at an unprecedented pace on a global scale (*Willis et al., 2017*). Between 4 and 12 million metric tons of land-based plastic waste is estimated to have entered the marine environment in 2010 alone (*Jambeck et al., 2015*), and the forecast is for a cumulative increase to 12 billion metric tons by the year 2050 (*Geyer, Jambeck & Law, 2017*). Common plastics accumulate in landfills and the natural environment because they decompose very slowly under natural conditions (*Barnes et al., 2009*). This is why the almost permanent pollution of the natural environment by plastic waste is becoming more and more of a problem. *Cole et al. (2011)* stated that plastics account for a significant portion of marine litter. The UV rays of sunlight promote the degradation of plastic waste into small fragments called microplastics (MPs, <5 mm) (*Browne, Galloway & Thompson, 2007*; *Andrady, 2011*), and MPs have been found in major oceans and coastal areas (*Barnes et al., 2009*).

Corresponding author
Hiroshi Asakura,
asakura_hiroshi@yahoo.co.jp

MPs have the potential to accumulate organic pollutants such as carcinogenic polychlorinated biphenyls (PCBs) (*Mato et al., 2001*; *Frias, Sobral & Ferreira, 2010*; *Bellas et al., 2016*), polycyclic aromatic hydrocarbons (PAHs), and others (*Rochman et al., 2012*; *Rochman et al., 2013*), which eventually results in the organic pollutants entering the marine food web (*Vandermeersch et al., 2015*). Plastics were detected in the gastrointestinal tracts of 36.5% of fish in the English Channel (*Lusher, McHugh & Thompson, 2013*). Polybrominated diphenyl ethers were detected in plastic fragments found in the stomachs and abdominal adipose tissues of oceanic seabirds (*Tanaka et al., 2012*).

Measures to prevent marine pollution and reduce the use of plastics are being taken worldwide, including Sustainable Development Goals (SDGs) adopted in 2015 and "A European Strategy for Plastics in a Circular Economy" released in 2018. As these measures take effect, the amount of MPs in the environment will probably decrease in the long term, although it may be necessary to collect existing plastic wastes. To evaluate the effectiveness of the measures, we must monitor trends in MPs abundance.

MPs in marine sediment such as sand have been intensively investigated (*Ng & Obbard, 2006*; *Corcoran, Biesinger & Grifi, 2009*; *Frias, Sobral & Ferreira, 2010*; *Turner & Holmes, 2011*; *Martins & Sobral, 2011*; *Hidalgo-Ruz & Thiel, 2013*; *Mathalon & Hill, 2014*; *Masura et al., 2015*; *Wessel et al., 2016*; *Crichton et al., 2017*; *Maes et al., 2017*; *Quinn, Murphy & Ewins, 2017*). However, those studies use various approaches to identify, quantify, and report measured concentrations of MPs, making spatiotemporal comparisons difficult (*Hidalgo-Ruz et al., 2012*). To address this issue, attempts have been made to standardize the investigation method for beach litter (UNEP Guidelines (*Cheshire et al., 2009*), OSPAR (*Wenneker & Oosterbaan, 2010*), and NOAA (*Burgess et al., 2021*)). In addition, a few methods for investigating beach litter, including MPs, have been developed (JRC *Galgani et al., 2013*) and GESAMP (*Kershaw, Turra & Galgani, 2019*).

The author has some doubts about the way the results of MPs investigation are represented. First, the author think the MPs present in our immediate environment are not the essence of long-term environmental pollution trends. Environmental pollution by MPs is caused by the presence of large amounts of MPs in a given space. This is expressed as the MPs content or abundance, *i.e.,* the number or weight of MPs per size or weight of the medium, such as sand. However, MPs abundance is a secondary phenomenon. The presence of MPs in a given space is due to the arrival of MPs from a source. Moreover, it is the rate of accumulation that determines the amount of MPs present after very long periods of time and after beach cleanings. In other words, the inflow and accumulation of MPs is a primary phenomenon and the essence of long-term MPs environmental pollution. Therefore, in this study, we measure the accumulation rate of MPs. The aforementioned guidelines have also recommended the investigation of the accumulation rate to determine long-term trends in environmental contamination of MPs. JRC (*Galgani et al., 2013*) states, "only the accumulation surveys provide information on the rate of deposition of litter and trends in litter pollution". GESAMP (*Kershaw, Turra & Galgani, 2019*) states, "A common goal for marine litter monitoring surveys is to address specific policy-related questions", and "Typical questions might include: is the total amount of marine litter on the shoreline increasing or decreasing? The best way to answer such questions is to conduct accumulation

surveys". While reports on the MPs abundance in coastal areas are abundant, reports on the accumulation rate are scarce. Several reports on the accumulation rate of marine litter other than MPs have been reported, for example, by *Eriksson et al. (2013)*, *Lee & Sanders (2015)*, and *Dunlop & Dunlop (2020)*. The rate of accumulation of MPs on coastal area is not well understood.

Next, the MPs content determined by a fixed or hot spot (HS) investigation in a certain sandy beach does not represent the MPs content of the entire beach. When we compare the MPs content of multiple beaches or evaluate changes in MPs content over time, we should be targeting entire beaches or portions of them, not specific survey points. First, if the fixed spot happens to be an HS or a non-hot spot (nHS), the observed value cannot be considered a representative value. To resolve this issue, the number of samples can be increased, but this is not realistic because labor is increased as well. Second, the reported MPs content of an HS in a certain sandy beach is the measurement value for that part of the beach that has a particularly high MPs content, not the representative value of the entire beach. JRC (*Galgani et al., 2013*) and GESAMP (*Kershaw, Turra & Galgani, 2019*) recommend that samples be taken at the strandline. The strandline is a part of HS because of the accumulation of marine litter. Many studies used strandline areas, the HS, to evaluate the MPs abundance (*Barnes et al., 2009*; *Browne et al., 2011*; *Martins & Sobral, 2011*; *Hidalgo-Ruz & Thiel, 2013*; *Dekiff et al., 2014*). We wish to compare the MPs contents of entire sandy beaches, but are actually comparing the selected and highest MPs values. In other words, "We are asked about the population density of a country as a whole, but we answered the population density of a city". On the other hand, *De-la Torre et al. (2020)* sampled sand in several transects to compare MPs abundance on four beaches in Lima, Peru, and found large variability. The importance of understanding distribution and variability has been noted for estimating MPs abundance on sandy beaches. For example, *Moreira et al. (2016)* investigated small-scale temporal and spatial variability in the state of Paraná, southern Brazil, to provide comparable MPs abundance estimates across areas. *Leads et al. (2023)* studied spatio-temporal variation in South Carolina, in the southeastern United States, to accurately understand the level of MPs contamination in the environment. Thus, the need for estimates of MPs abundance that are comparable across areas is noted. To estimate representative values of MPs content for the entire study area, observations can be weighted by frequency, *e.g.*, area.

We propose a method to represent the results of investigation of MPs abundance in sand of sandy beaches. The questions to be answered are as follows: what would be the magnitude of the accumulation rate of MPs in the study site? Would it be possible to estimate the amount of MPs present in the entire study site? To answer these questions, the MPs accumulation rate in the study site was measured by periodic investigation at fixed spots. In addition, the weighting of the MPs content in hot spots and non-hot spots by their respective areas to obtain the average and error (mg-MPs/m$^2$-sand) of the MPs content in the entire study site. This study provides a rare example of measuring the MPs accumulation rate on the coastal area and proposes a method for making significant regional comparisons of MPs abundance.
## MATERIALS & METHODS

In this section, the overall process in the MPs investigation method is disclosed. The methods used in the fixed spot investigation to determine the MPs accumulation rate and the HS investigation to determine the average and error of MPs present in the entire study site are also described.

### Overall process

First, the names of the locations where the MPs investigation will be conducted are defined. A somewhat large area to be studied, such as a sandy beach, is called the "study site". At the study site, one unit block where sand is sampled is called the "sampling square", which corresponds to the quadrat in GESAMP (*Kershaw, Turra & Galgani, 2019*).

Second, the study site is determined. In this study, the investigation is conducted in Nagasaki Prefecture located on Kyushu Island in western Japan. Sandy beaches A (32.7713, 129.8017), B (33.3566, 129.5007), C (34.2354, 129.1915), and D (32.7443, 128.6933) in the prefecture are selected as the study sites. Field experiments were approved by Nagasaki Prefecture (project number: 2022-1). Fixed spot investigations are conducted at Beach A, and HS investigations are conducted at Beaches B, C, and D. The investigation periods are: Beach A: July 2021 to December 2022 (6 times), Beach B: September 2022, Beach C: October 2022, and Beach D: December 2022.

A flow chart of the investigation procedure is shown in Fig. 1. First, the sampling square (1 m × 1 m) is determined. In the case of fixed spot investigation, the sampling square is determined by land survey. In the case of HS investigation, the sampling square is visually determined first, and then its location is recorded by land survey. Surface sand (1 cm depth) from the sampling square is collected and flotation sorting using seawater is performed following the method described by *Asakura (2023)*. Floating matter brought back to the laboratory contains particles other than MPs. MPs are sorted from the floating matter, washed with tap water, dried, visually sorted, and weighed using the method described by *Asakura (2023)* (Fig. 1). This study employs an investigation method that only covers a limited number of MPs with particle sizes of 1–5 mm and densities of less than 1 g/cm$^3$. The lower limit of quantification for MPs analysis in this study is 13 mg-MPs/m$^2$-sand (*Asakura, 2023*).

Aluminum scoop (AL250; Wilesco, Lüdenscheid, Germany), stainless steel sieve ($\varphi$4.75 mm; Sanpo, Taiwan), handy scale (LS-50; Custom), electric dryer (ADVANTEC, DRD420DA; Advantec, Taipei City, China), electronic scale (ATY124; Shimadzu, Tokyo, Japan) were used as the main instruments for MPs analysis.

JRC (*Galgani et al., 2013*) recommends particle count as the standard unit of MPs. *Frias et al. (2018)* recommend that MPs be reported by weight as well as by number. The advantage of particle count is that it can report the presence of very small particles. The disadvantage is that when a particle is split in two, the number of particles doubles while the amount of plastic present remains constant. The advantage of weight measurement is that the amount of plastic itself can be reported. The disadvantage is that when the particles are very small, it is impossible to measure them because they are below the lower limit of quantitation of the electronic balance. In this study, we report weights in order to eliminate

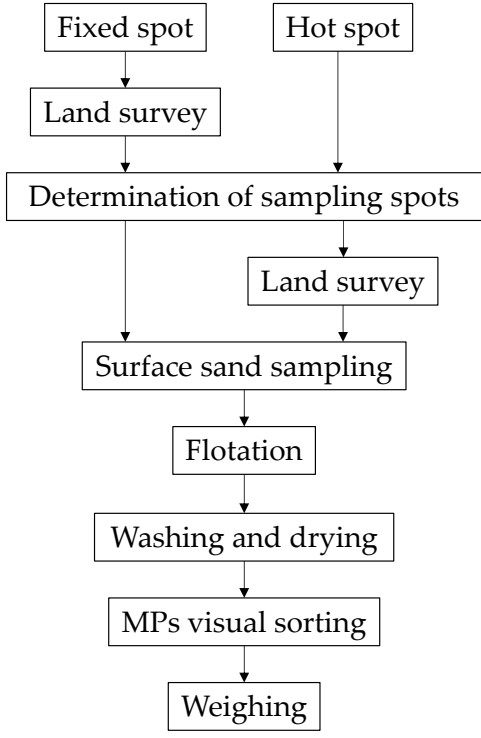

**Figure 1 Flow chart of procedure for investigating MPs content.**

concerns about changes in measurements due to particle destruction when comparing the abundance of MPs at study sites. Note that the author also possesses data on the number of particles.

## Calculation of MPs accumulation rate by fixed spot investigation

Suppose we have one-time MPs measurement results for several study sites. Beach X is 1,000 mg-MPs/m$^2$-sand, and Beach Y is 3,000 mg-MPs/m$^2$-sand. At first glance, it appears that the Beach Y has approximately three times more serious MPs pollution than Beach X. However, these findings only represent the MPs content at that moment (Fig. 2A). The importance of MPs pollution may lie not so much in the MPs present at a given moment as in the MPs drift rate (accumulation rate) that causes it. This is because it is sandy beaches with larger accumulation rates that are more contaminated after beach cleaning. Furthermore, even if there is no beach cleaning, assuming an infinite number of elapsed days, sandy beaches with larger accumulation rates will be more contaminated. Assuming that Beach X and Beach Y have been contaminated from a clean state 10 days and over 300 days, respectively, MPs contamination at Beach X is more serious than that at Beach Y. If we simply calculate the accumulation rate in this case, Beach X will be 100 mg-MPs/(m$^2$-sand·d), and Beach Y will be 10 mg-MPs/(m$^2$-sand·d). In other words, MPs pollution in Beach X is approximately 10 times more serious than that in Beach Y. JRC (*Galgani et al., 2013*) and GESAMP (*Kershaw, Turra & Galgani, 2019*) have

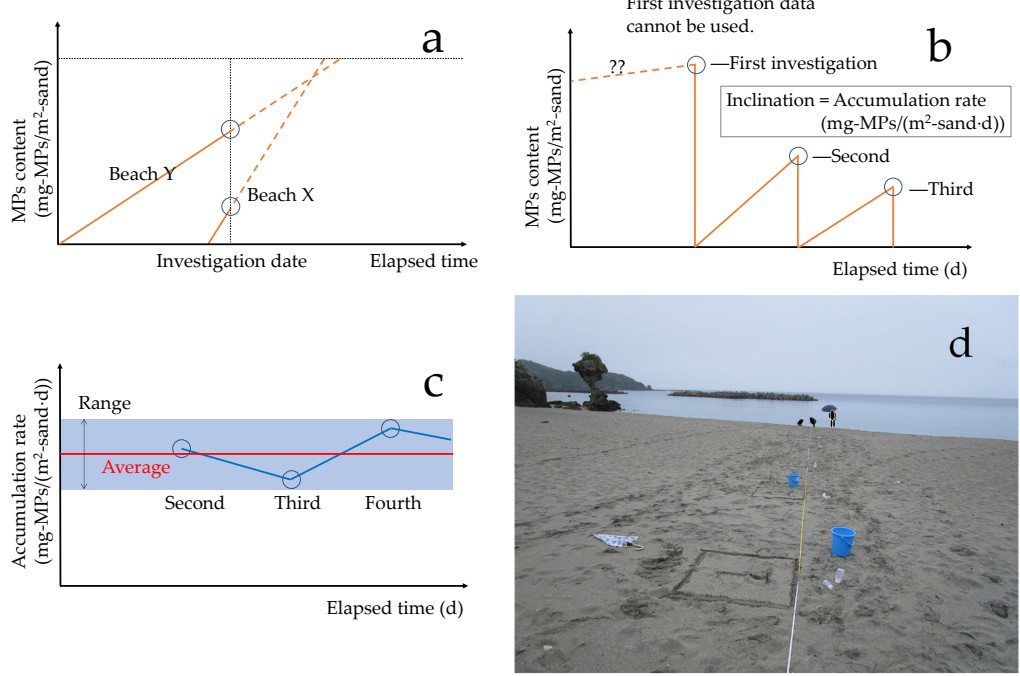

**Figure 2 MPs measurement at fixed spots and accumulation rate.** (A) One-time MPs measurement results (circle) and accumulation rates (inclination); (B) Calculation of MPs accumulation rate by repeated measurements at a fixed spot; (C) Obtained multiple MPs accumulation rates; (D) MPs measurement at a fixed spot.

also recommended the investigation of the accumulation rate to determine the trend of pollution by marine litter.

In this study, the MPs on the sampling square of a study site visited for the first time ($i = 1$) are collected and brought back to the laboratory to be measured ($C_{i=1}$). Therefore, the MPs on the sampling square are considered to have "disappeared" because they have been collected. After a certain period of time ($t_i$), MPs are collected from the same sampling square (fixed spot) and brought back for measurement ($C_i$). The accumulation rate $r_{acc i}$ (mg-MPs/(m²-sand·d)) is:

$$r_{acc i} = C_i/t_i. \tag{1}$$

The first measurement $C_{i=1}$ is discarded (Fig. 2B). In this study, the MPs accumulation rate is calculated by repeatedly measuring MPs at several fixed spots on Beach A after a certain period of time (Figs. 2C and 2D). A single MPs accumulation rate is not reliable. By obtaining multiple MPs accumulation rates, the representative value and the dispersion would be obtained. MPs are collected and measured at distances of 4, 8, 12, 16, 20, and 40 m from the concrete road to the sea, as shown in Fig. 3A. Point 4 m is closest to the road and the highest elevation; points 4 to 12 m are in the supratidal zone and are dry; and points 16 and 40 m are in the intertidal zone and are wet. How to establish appropriate fixed spots for estimating the average accumulation rate in a study site is an important
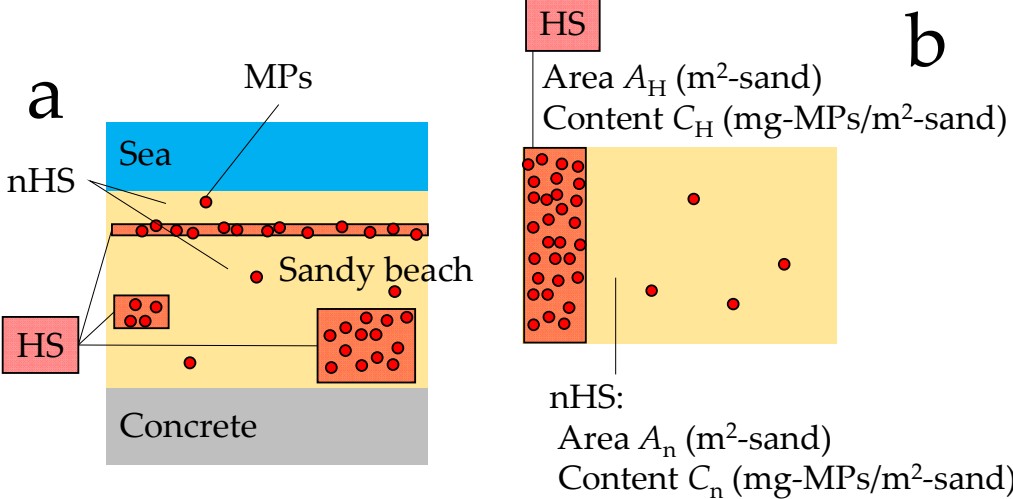

**Figure 3 Hot and non-hot spots (HS and nHS, respectively).** (A) Conceptual diagram of hot and non-hot spots on a beach; (B) Simplified hot and non-hot spots, and their areas and MPs contents.

topic. However, in this study, fixed spots were simply set near the center of Beach A as a basic study.

## Representation of MPs presence in entire study site by hot spot investigation

By imagining the phenomenon of MPs drifting on sandy beaches or gathering in blowholes because of wind and other factors, it can be expected that the MPs content will vary depending on the location. In this case, the method recommended by JRC (*Galgani et al., 2013*) and GESAMP (*Kershaw, Turra & Galgani, 2019*) of measuring MPs content at locations such as the strandline where it can be visually confirmed that there is an abundance of MPs, and using the measurement results as the representative value and the dispersion of MPs content for the entire study site would be lacking in accuracy. The method would probably yield a higher value than the true representative value. Thus, a method that also targets areas with low MPs content should be considered as it will provide high and low values of MPs content. However, because these values cannot be weighted, a simple average cannot be used as the representative value.

Therefore, in a certain beach, the areas where the presence and absence of MPs can be visually confirmed are considered as HS and nHS, respectively, the MPs content in each spot is measured at several locations (sampling squares; $i = 1, 2, \ldots, n$), and the average value is obtained. The area of each compartment is also measured, and the MPs content is weighted by the area ratio. From the above, we express the representative value and the dispersion of the MPs content in the entire sandy beach. Figure 3A shows a conceptual diagram of HS and nHS in a sandy beach. Figure 3B shows the simplified HS and nHS, and their areas and MPs contents.

Subscripts H and n denote HS and nHS, respectively. Let HS area denote $A_H$ (m²-sand) and nHS area, $A_n$ (m²-sand) on a given beach. In this case, the area ratio $r$ of each spot is:

$$r_H = A_H/(A_H + A_n) \tag{2}$$

$$r_n = A_n/(A_H + A_n). \tag{3}$$

If the sample size is $n$ and the individual MPs content in sampling square is $C_i$ (mg-MPs/m²-sand), the average MPs content $C_{\_ave}$ (mg-MPs/m²-sand) is:

$$C_{H\_ave} = (1/n_H)\Sigma C_{Hi} \tag{4}$$

$$C_{n\_ave} = (1/n_n)\Sigma C_{ni}. \tag{5}$$

The variability of measurements is expressed by the following sample variance $s^2$:

$$s_H^2 = \{1/(n_H - 1)\}\Sigma(C_{Hi} - C_{H\_ave})^2 \tag{6}$$

$$s_n^2 = \{1/(n_n - 1)\}\Sigma(C_{ni} - C_{n\_ave})^2 \tag{7}$$

Then, the confidence interval for the population mean of the MPs content is determined. Let the confidence coefficient be $1 - \alpha$. The estimator of the population mean is equal to $C_{\_ave}$. The confidence interval for the population mean can be obtained using the $t$-distribution as:

$$C_{H\_ave} \pm t_{\alpha/2}(n_H - 1)s_H/\sqrt{n_H} = C_{H\_ave} \pm \delta C_{H\_ave} \tag{8}$$

$$C_{n\_ave} \pm t_{\alpha/2}(n_n - 1)s_n/\sqrt{n_n} = C_{n\_ave} \pm \delta C_{n\_ave} \tag{9}$$

The second term $\delta C_{\_ave}$ is called the error.

We want to express the MPs content (mg-MPs/m²-sand) of the entire sandy beach using the average and the error. We aggregate (synthesize) HS and nHS into a total area of 1 m² while maintaining the MPs content and area ratio. The average and the error are synthesized, respectively.

The average content of the entire beach, $C_{all\_ave}$ (mg-MPs/m²-sand), is obtained by weighting the average content of each plot by the area ratio:

$$C_{all\_ave} = r_H C_{H\_ave} + r_n C_{n\_ave} \tag{10}$$

Next, we consider the synthesized error $\delta C_{all\_ave}$ (mg-MPs/m²-sand). Equation (10) shows that the measured values (average values) are multiplied by a constant area ratio and then summed over these measured values. Therefore, considering the propagation of errors associated with the measured values (average values), we obtain:

$$\delta C_{all\_ave} = \sqrt{\{(\partial C_{all\_ave}/\partial C_{H\_ave})^2(\delta C_{H\_ave})^2 + (\partial C_{all\_ave}/\partial C_{n\_ave})^2(\delta C_{n\_ave})^2\}}$$
$$= \sqrt{\{r_H^2(\delta C_{H\_ave})^2 + r_n^2(\delta C_{n\_ave})^2\}}. \tag{11}$$

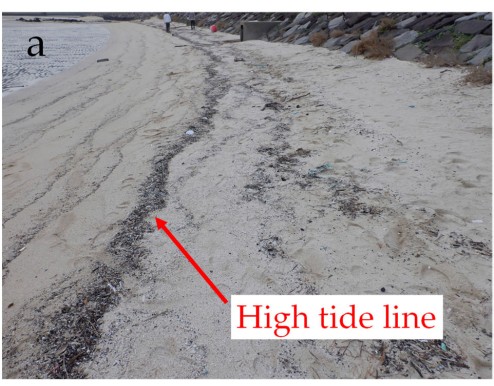
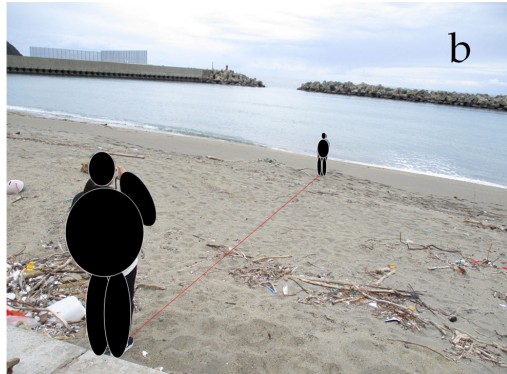

High tide line

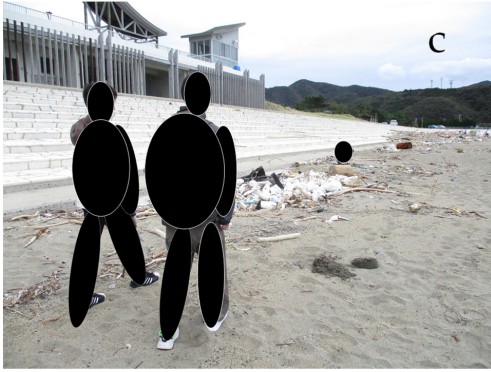

**Figure 4 Hot spot and non-hot spot investigation.** (A) High tide line; (B) The distance between artifact and high tide line is measured with a laser rangefinder; (C) Hot spots are visually determined and their areas and locations recorded.

From the above, the MPs content (mg-MPs/m$^2$-sand) of the entire beach can be expressed using the average and the error as follows:

$$C_{\text{all\_ave}} \pm \delta C_{\text{all\_ave}}. \tag{12}$$

HS investigations were conducted at Beaches B, C, and D. The beach was considered to extend from the high tide line to the artifacts on the land side (roads, stairs, or embankments). The high tide line was clearly identified visually because it was littered with debris (Fig. 4A). The distance from the artifacts to the high tide line was measured with a laser rangefinder or a tape measure (Fig. 4B). The HS was visually determined, and its area and location were recorded. Because of the large number of HSs and their large size, surveying them with a tape measure was cumbersome. Therefore, we trained 10 times to make 20 steps over a distance of 10 m (2 steps/m) and surveyed the area and location by the number of steps (Fig. 4C). HS and nHS were relatively easy to identify visually (Fig. 5). For each beach, 12 sampling squares of HS ($n = 12$) and nHS ($n = 12$) were extracted and determined. The sampling squares were distributed throughout the study site. The location of the sampling square was surveyed using the method described by *Asakura (2023)*. MPs from the sampling squares were collected and brought back to the laboratory.

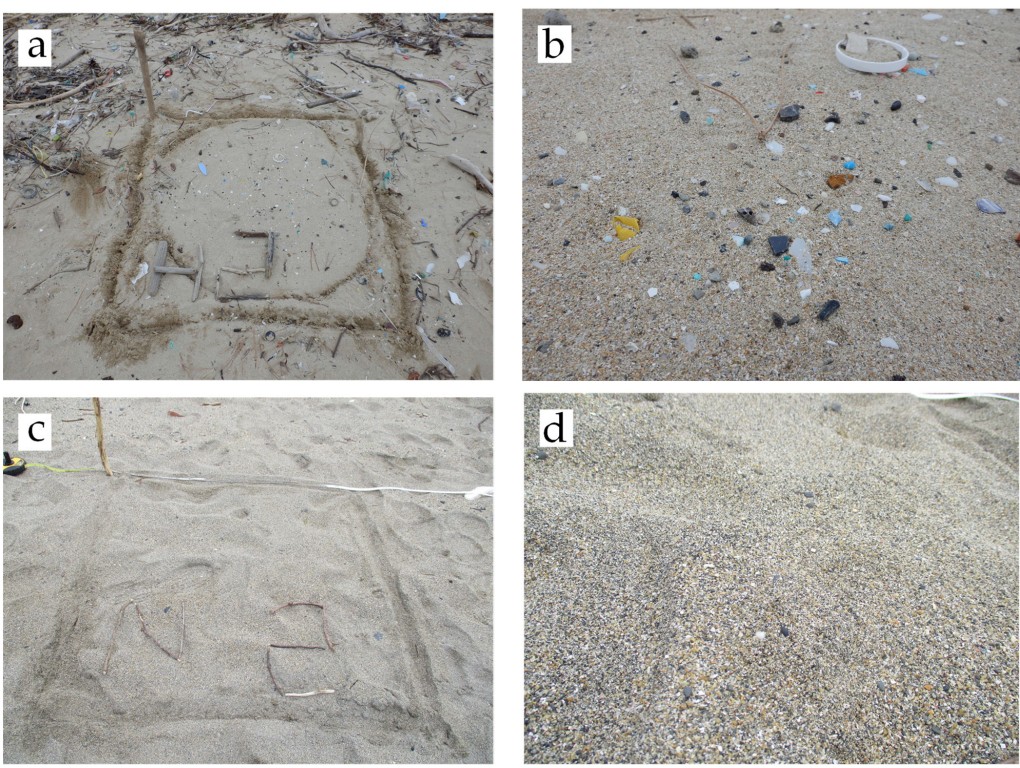

**Figure 5** **Photographs of hot and non-hot spots.** (A) Hot spot (sampling square); (B) Hot spot (close-up); 16,000 mg-MPs/$m^2$-sand; (C) Non-hot spot (sampling square); (D) Non-hot spot (close-up); 26 mg-MPs/$m^2$-sand.

After returning to the laboratory, MPs were measured and the survey data were plotted on an aerial photograph on Google Maps. First, the high tide line was plotted (Fig. 6A). The radius of the green circle is the distance from an artifact to the high tide line. In the area between the high tide line and the artifact (backshore), squares and triangles were laid out and their areas were summed to obtain the total area of the backshore. For the calculation of the area, scale = distance on the screen/actual distance was needed. Next, the HS was drawn and the area $A_H$ was also obtained (Fig. 6B). Although a rough estimate, we considered all areas other than the HS as nHS, and subtracted the HS area from the total area of the backshore to obtain the nHS area $A_n$. The sampling squares were drawn using the method described by *Asakura (2023)*, and the MPs content was expressed as the area of the circle (for example, Figs. 6C and 6D).

## Personnel and time required for MPs investigation

Continuous investigations are necessary to determine the actual status of environmental pollution by MPs. When attempting to express MPs abundance in the entire study site using average and error, the inaccuracy will increase and the information obtained will be worthless unless a sufficiently large sample size is collected. In other words, an unplanned visit to the study site may end up with meaningless results. To collect a sufficiently large

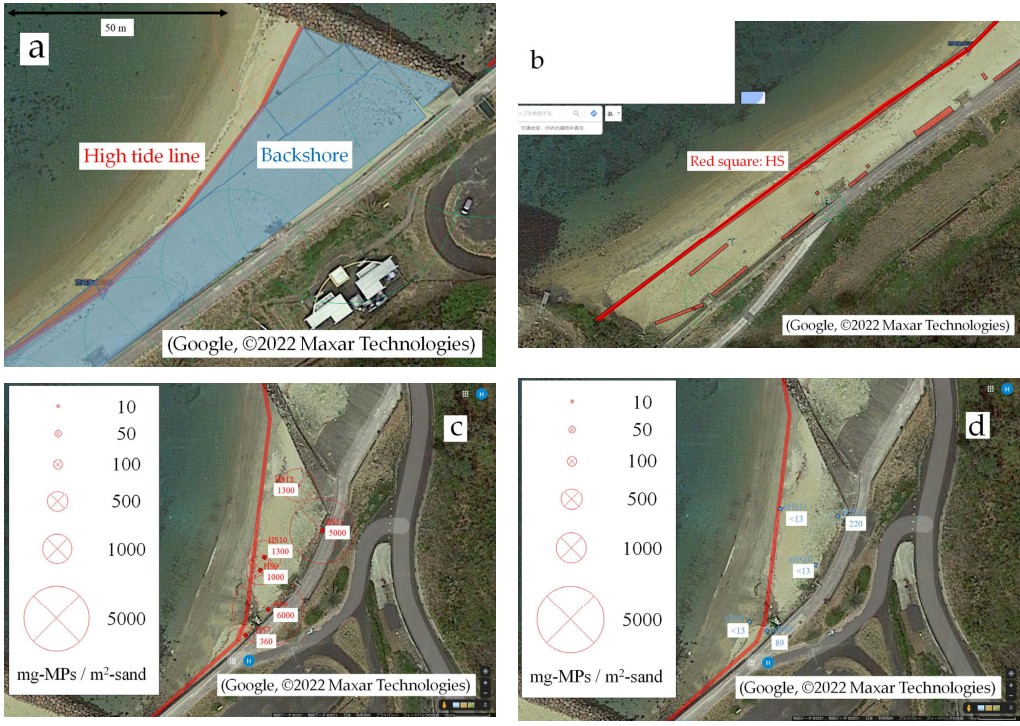

**Figure 6  Drawing of measurement results on aerial photographs.** (A) The high tide line was plotted, and the backshore area was estimated; (B) Hot spots were drawn and their areas estimated; (C) Bubble chart of MPs content in hot spots; (D) Bubble chart of MPs content in non-hot spots. Image source credits: Google, ©2022 Maxar Technologies.

number of samples, sufficient labor and time are needed. As reference for the development of a successful investigation plan, the personnel and time required for MPs investigation, which are determined on the basis of experience gained in this study, are shown in Table 1.

## Data analysis
A one-way analysis of variance was performed to compare MPs accumulation rates. Multiple comparisons using the Tukey method were performed to compare MPs abundance in entire study site. Logistic regression analysis was performed to determine the Relationship between MPs content and visual HS or nHS determination. For statistical processing, Data Analysis Toolpak in Microsoft Excel 2019 (Microsoft, Redmond, WA, USA) was used.

## RESULTS
### Fixed spot investigation
Table 2 shows the MPs contents in fixed spots at Beach A. Because the fixed spot investigation did not select spots where MPs were abundant, many of the measurement results were below the lower limit of quantification. The points 16 to 40 m from the road were intertidal, and MPs were pushed toward the landward side, so MPs were considered

**Table 1  Personnel and time required for MPs investigation.**

|  | Unit | Time | Time | Minimum number of persons |
|---|---|---|---|---|
| Preparation | min·p | 15 |  | 1 |
| HS decision | min·p/spot | 5[b] | 3[c] | 1 |
| nHS decision | min·p/spot | 2 |  | 1 |
| Fixed spot survey | min·p/spot | 3 min × 3p |  | 3 |
| n/HS survey | min·p/spot | 5 min × 3p | 2 min × 3p[d] | 3 |
| Collection of floating matter in bucket[a] | min·p/spot | 15[b] | 20[c] | 1 |
| Drawing water | min·p/bkt | 1[e] |  | 1 |
| Finishing | min·p | 15 |  | 1 |

Notes.
[a] Sand sampling, sand weighing, addition into water and agitation, photography, and collection of floating matter
[b] Small amount of MPs
[c] Large amount of MPs
[d] When the fixed first tape measure can be reused
[e] It takes time to make preparations for drawing water, but once drawing started, it can be collectively done quickly. The expression "5 min·p/5 bkt" is appropriate.
p, person; bkt, bucket.

**Table 2  MPs contents in fixed spots at Beach A.**

| Date | | 27-Jul-21 | 22-Sep-21 | 24-Nov-21 | 11-May-22 | 24-Aug-22 | 01-Nov-22 |
| Season | | Summer | Autumn | Winter | Spring | Summer | Autumn |
| Number $i$ | | 1 | 2 | 3 | 4 | 5 | 6 |
| Elapsed time (d) | | 0 | 57 | 120 | 288 | 393 | 462 |
|---|---|---|---|---|---|---|---|
| Distance from | 4 | 127 | 85 | 45 | 68 | 28 | 203 |
| road (m) | 8 | <13 | 112 | 279 | <13 | 751 | 94 |
|  | 12 | <13 | 15 | 20 | <13 | 52 | 23 |
|  | 16 | <13 | 142 | <13 | <13 | 18 | <13 |
|  | 20 | <13 | <13 | <13 | 17 | <13 | <13 |
|  | 40 | – | – | – | <13 | <13 | <13 |
|  |  |  |  |  |  |  | mg-MPs/m$^2$-sand |

absent. The frequency of the measurement results that were above the lower limit of quantification was higher on the landward side than on the seaward side.

## Hot spot investigation

Table 3 and Fig. 7 show the MPs contents in HS and nHS at Beaches B, C, and D. MPs content was approximately two orders of magnitude lower in nHS than in HS. In HS at Beach D, there was a clear division between high MPs content and relatively low MPs content, although they were in the same HS category. In nHS, some MPs contents were near the lower limit of quantification, while some others were comparable to the MPs contents in HS. For example, for Beach D, the highest value in nHS was 440 mg-MPs/m$^2$-sand and the lowest value in HS was 430 mg-MPs/m$^2$-sand (Fig. 7).

**Table 3** MPs contents (average (Ave) and standard deviation (SD)) in hot spot (HS) and non-hot spot (nHS) at Beaches B, C, and D ($n = 12$ for HS and 12 for nHS).

| | B | | C | | D | |
|---|---|---|---|---|---|---|
| | Ave | SD | Ave | SD | Ave | SD |
| HS | 2,260 | 1,992 | 4,002 | 2,975 | 12,138 | 17,611 |
| nHS | 46 | 63 | 25 | 19 | 101 | 125 |
| | | | | | mg-MPs/m²-sand | |

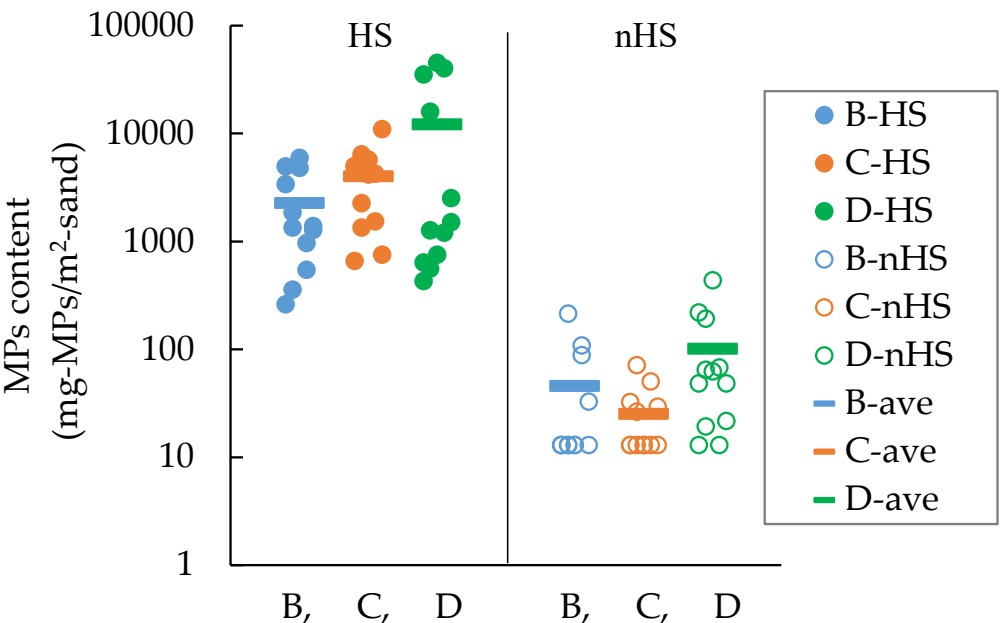

**Figure 7** MPs contents in hot spot (HS) and non-hot spot (nHS) at Beaches B, C, and D ($n = 12$ for HS and 12 for nHS).

## DISCUSSION

### MPs accumulation rate

The accumulation rates of MPs for the fixed spot investigation at Beach A are shown in Fig. 8. Accumulation rates for 4, 8, 12 m ($n = 5$) and all ($n = 15$) in the supratidal zone were $1.2 \pm 1.1$, $3.0 \pm 2.7$, $0.3 \pm 0.1$, and $1.5 \pm 0.9$ mg-MPs/(m²-sand·d) (average $\pm t_{0.05/2}(n-1)$ · standard error). One-way analysis of variance for 4, 8, and 12 m showed no significant difference in accumulation rates ($p = 0.11$, $n = 15$). Accumulation rate in sandy beach A varies depending on the sampling point, but seems to be a few mg-MPs/(m²-sand·d) on average. Continued investigations are needed to determine the long-term characteristics of the MPs accumulation rate.

Few reports on the accumulation rate of MPs on sandy beaches have been published. MPs accumulation rates of 0.18 to 1.61 kg/(8,000 m² ·30 days), *i.e.,* 0.75 to 6.7 mg/(m² ·d) were obtained in marshes and beaches in Georgia, USA (*Lee & Sanders, 2015*), and those values are similar to the accumulation rates obtained in this study.

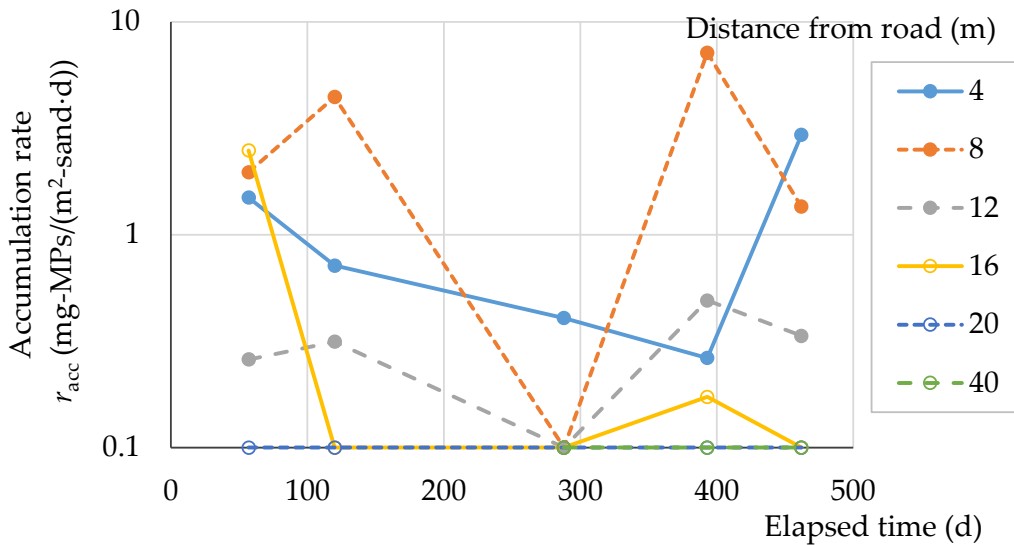

**Figure 8  MPs accumulation rate at Beach A.** 4 m: the highest elevation at backshore, 4–16 m: backshore (dry), 20–40 m: intertidal (wet).

*Eriksson et al. (2013)* studied plastic fragments drifting per day on a 1 km wide beach. *Dunlop & Dunlop (2020)* studied daily accumulation rates of marine litter on beach. Some examples of measurements of the accumulation rates in bottom sediments in water are cited (*Turner et al., 2019*; *Rios-Mendoza et al., 2021*; *Saarni et al., 2021*; *Hinata et al., 2023*). *Yuan et al. (2023)* measured the annual deposition flux from the atmosphere. We hope to find more studies showing similar MPs accumulation rates to that reported in this study in order to allow for comparisons among multiple sandy beaches.

## Representation of MPs presence in entire study site

Table 4 shows the weighted results of MPs investigation for Beaches B, C, and D. The average is shown at the bottom row of the column "Weighted average", and the error is shown at the bottom of the rightmost column ($\alpha = 0.05$). MPs content for Beach D is 4,090 ± 3,709 mg-MPs/m$^2$-sand, and the error is extremely large (D-1). The MPs abundance in HS clearly differs at the boundary of a certain point in Beach D. This can be seen from the fact that the MPs contents are clearly divided, with one group lying above the average value and another group, below the average value of HS at Beach D in Fig. 7. Therefore, HS at Beach D is divided into HS1 (high) and HS2 (low), and the MPs content is 4,084 ± 2,243 mg-MPs/m$^2$-sand as shown in Table 3 (D-2); the error is smaller than that for D-1. To reduce errors, it may be necessary to divide the HS into several categories according to abundance of MPs. However, changing the calculation procedure after the results have been obtained without a valid reason is equivalent to statistical fraud. As in the present case, it is necessary to have evidence such as different concentration distributions for each block. Table 5 shows the MPs content per area for Beaches B, C, and D. Shown as average ± error, the MPs contents were 298 ± 144 mg-MPs/m$^2$-sand, 1,115 ± 518 mg-MPs/m$^2$-sand, and 4,084 ± 2,243 mg-MPs/m$^2$-sand, respectively. The

**Table 4 Weighted results of MPs investigation ($n = 12$ for HS and 12 for nHS).**

**Beach B**

| | Average (a) $C_{ave}$ mg-MPs/m²-sand | Error (b) $\delta C_{ave}$ mg-MPs/m²-sand | Area $A$ m²-sand | Area ratio (c) $r$ – | Weighted average $a \times c$ mg-MPs/m²-sand | Error calculation $c^2 \times b^2$ – |
|---|---|---|---|---|---|---|
| HS | 2,260 | 1,266 | 533 | 0.11 | 257 | 20,776 |
| nHS | 46 | 40 | 4,145 | 0.89 | 41 | 31 |
| Total | | | 4,677 | | 298 | 20,808 |
| Total error | $\delta C_{all\_ave}$ | mg-MPs/m²-sand | | | | 144 |

**Beach C**

| | Average (a) $C_{ave}$ mg-MPs/m²-sand | Error (b) $\delta C_{ave}$ mg-MPs/m²-sand | Area $A$ m²-sand | Area ratio (c) $r$ – | Weighted average $a \times c$ mg-MPs/m²-sand | Error calculation $c^2 \times b^2$ – |
|---|---|---|---|---|---|---|
| HS | 4,002 | 1,890 | 409 | 0.27 | 1,097 | 268,518 |
| nHS | 25 | 12 | 1,082 | 0.73 | 18 | 6 |
| Total | | | 1,491 | | 1,115 | 268,525 |
| Total error | $\delta C_{all\_ave}$ | mg-MPs/m²-sand | | | | 518 |

**Beach D-1**

| | Average (a) $C_{ave}$ mg-MPs/m²-sand | Error (b) $\delta C_{ave}$ mg-MPs/m²-sand | Area $A$ m²-sand | Area ratio (c) $r$ – | Weighted average $a \times c$ mg-MPs/m²-sand | Error calculation $c^2 \times b^2$ – |
|---|---|---|---|---|---|---|
| HS | 12,138 | 11,189 | 347 | 0.33 | 4,023 | 13,755,289 |
| nHS | 101 | 80 | 700 | 0.67 | 67 | 36 |
| Total | | | 1,047 | | 4,090 | 13,755,325 |
| Total error | $\delta C_{all\_ave}$ | mg-MPs/m²-sand | | | | 3,709 |

**Beach D-2**

| | Average (a) $C_{ave}$ mg-MPs/m²-sand | Error (b) $\delta C_{ave}$ mg-MPs/m²-sand | Area $A$ m²-sand | Area ratio (c) $r$ – | Weighted average $a \times c$ mg-MPs/m²-sand | Error calculation $c^2 \times b^2$ – |
|---|---|---|---|---|---|---|
| HS1 | 34,191 | 20,340 | 116 | 0.11 | 3,771 | 5,032,926 |
| HS2 | 1,111 | 574 | 232 | 0.22 | 246 | 28 |
| nHS | 101 | 80 | 700 | 0.67 | 67 | 36 |
| Total | | | 1,047 | | 4,084 | 5,032,990 |
| Total error | $\delta C_{all\_ave}$ | mg-MPs/m²-sand | | | | 2,243 |

area of HS ranges from 11 to 33% of the total area. Multiple comparisons using the Tukey method showed significant differences between B–D and C–D ($p < 0.05$), but not between B–C.

Few measurements of MPs on sandy beaches have been reported in terms of weight per area (*i.e.*, mg/m²). In India, MPs range from approximately 300 to 1,500 mg/m² in a high tide line (*Jayasiri, Purushothaman & Vennila, 2013*) or 1,323 ± 1,228 mg/m² in a high tideline and 178 ± 261 mg/m² in a low tide line (*Karthik et al., 2018*). In Hong Kong, MPs of 5,600 mg/m² (*Fok & Cheung, 2015*) were found in a high strandline. Because these

**Table 5** MPs contents per area (average and error) for Beaches B, C, and D ($n = 24$).

|   | Average $C_{all\_ave}$ | Error $\delta C_{all\_ave}$ |
|---|---|---|
| B | 298 | 144 |
| C | 1,115 | 518 |
| D | 4,084 | 2,243 |
|   | mg-MPs/m²-sand | |

studies targeted mainly high tide line areas where MPs would be abundant, the measured values would be higher than the average value for the entire study site. The MPs contents measured in this study, which are averaged over the entire study site, are comparable to those reported in previous studies. In other words, MPs are present in high abundance in the entire study site examined in this study.

Because the sand collected from the sampling square in this study averaged 7 kg-sand/m²-sand, this factor can be used in the conversion into MPs content per weight of sand (mg-MPs/kg-sand). The MPs content per weight of sand from Beaches B, C, and D is approximately 40, 160, and 580 mg-MPs/kg-sand, respectively. In India, MPs content was 81 mg/kg at an intertidal zone (*Reddy et al., 2006*) and approximately 1 to 4 mg/kg at a transect along a shoreline (*Tiwari et al., 2019*); in Belgium, it was around 0.5 to 1 mg/kg at a high water mark (*Claessens et al., 2011*). As shown by *Asakura (2023)*, MPs are more abundant in the surface layer of sand than in the deep layers, that is, MPs content per weight of sand is larger when sand is sampled thinly, and smaller when sand is sampled deeply. In other words, the measured MPs content is dependent on the amount of sand, so care should be taken when referring to this value, *i.e.,* MPs content per weight of sand. The characteristics of the sampling methods are shown in Table 6. Since sampling at the strandline targets the HS, the measurement results are in principle higher than the true value. The transect method in Table 6 refers to the method of placing sampling points on grids. The random method refers to a method in which sampling points are randomly placed on the transect, or a method in which sampling points are randomly placed completely on the ground to be investigated. Both transect and random methods aim at unbiased sampling and can estimate the MPs abundance in the entire study site. However, when the sample size is small, the results are extremely variable depending on how many samples are selected from the HS. For example, in Beach C in Table 4, the MPs content in the HS is 160 (= 4,002/25) times larger than that in the nHS, and the area ratio is HS: nHS = 27: 73. On the other hand, in the weighting method by area proposed in this study, HS, which can determine the MPs abundance in the study site, is actively targeted for measurement and its frequency (area) is evaluated, so the variation in the overall estimate of the MPs abundance may be small. Naturally, this argument requires the assumption that the results of HS visual determination are correct.

### nHS is immediately adjacent to HS

In fixed spot investigations with sampling squares at regular distances, we expect to be able to estimate the population mean by those samples. However, it may be difficult to obtain

**Table 6** Sampling methods for estimating MPs abundance on coastal area.

| Method | Sampling point | Advantage | Disadvantage | Case example |
|---|---|---|---|---|
| Strandline | HS | Easy to determine the sampling points | Reported value higher than representative of the entire study site | 1, 2, 3, 4, 5 |
| Transect | Grid | Statistically justifiable procedures | Possibility of not extracting HS with small sample size | 6, 7, 8, 9, 10 |
| Randomized | Randomized | | | 11, 12, 13, 14, 15 |
| Weighting by area | HS and nHS | Evaluation of HS frequency | Error due to subjective HS determination | This study |

Notes.

HS, hot spot; nHS, non-hot spot [1] *Dekiff et al. (2014)*; [2] *Martins, Rodríguez & Pham (2020)*; [3] *Jocelyn et al. (2023)*; [4] *Azaaouaj et al. (2024)*; [5] *Sousa-Guedes et al. (2024)*; [6] *Nchimbi et al. (2022)*; [7] *Gül (2023)*; [8] *Kunz, Löwemark & Yang (2023)*; [9] *Bentaallah et al. (2024)*; [10] *Luan & Wang (2024)*; [11] *Moreira et al. (2016)*; [12] *Cruz-Salas et al. (2022)*; [13] *Leads et al. (2023)*; [14] *Lekshmi et al. (2023)*; [15] *Şener & Yabanlı (2023)*.

appropriate samples due to the extremely skewed spatial distribution of MPs content. Therefore, the author show that the spatial distribution of MPs content is extremely biased using the results of hot spot investigations, where the location of the sampling square can be determined by visual estimation of MPs content.

As shown in Fig. 7, there is a large difference in MPs content between HS and nHS. MPs were concentrated at HS in the field, and it was easy to find nHS at a distance of 1 m from the HS. This means that when sampling MPs in HS or nHS, the results can vary markedly if the sampling point is not set correctly. At the same time, it means that it is difficult to obtain appropriate samples for estimating the population mean in fixed spot investigations. To illustrate this, an nHS was set up in the immediate vicinity of an HS in pairs at Beach D. The contents ratio HS/nHS was divided by the distance to determine how many times the MPs content increased by 1 m distance (Fig. 9). In the area of HS1 where MPs content was particularly high, there were cases where the MPs content was 100 times ($n = 3$) or 700 times ($n = 1$) higher when the sampling point was shifted by 1 m.

*Kobayashi et al. (2021)* studied MPs in seawater in the East China Sea and reported a 550-fold difference between the highest and lowest abundances. This means that it is difficult to select a sampling point to obtain the representative value.

## Relationship between MPs content and visual HS or nHS determination

The fact that the visually determined HS actually contains more MPs than the nHS is a prerequisite for a hot spot investigation to be performed correctly. As shown in Fig. 7, the sampling squares with low MPs content were determined as nHS and those with high MPs content were determined as HS. Therefore, a logistic regression analysis was performed to create a model equation to perform this discrimination, and the discriminatory accuracy was determined. At the same time, the concentration at which the determination result switches can be known. Logistic regression analysis was conducted using the following (Eq. (13)), where $P$ is the probability that a sampling square determined to be HS, $x$ is the MPs content (mg-MPs/m$^2$-sand, $x = C_i$), and $b_0$ and $b_1$ are the partial regression coefficients.

$$P = 1/[1 + exp\{-(b_0 + b_1 x)\}]. \tag{13}$$

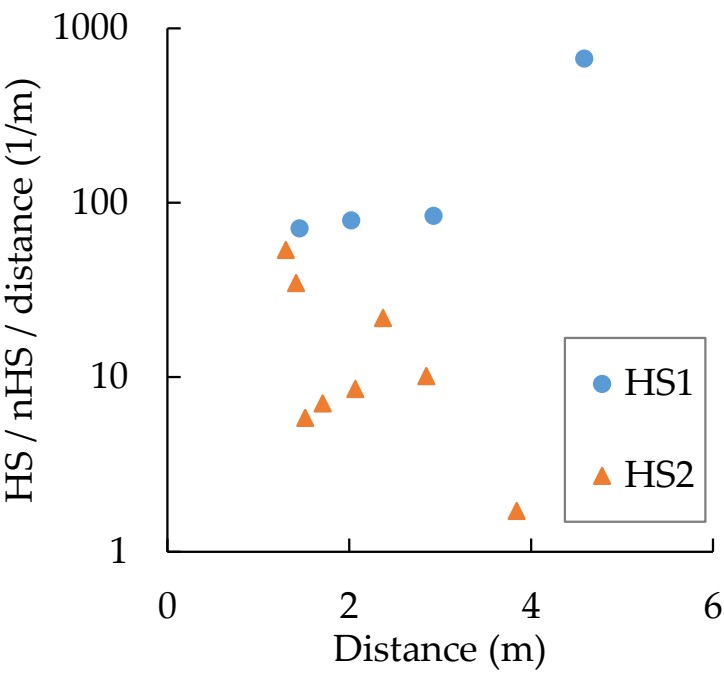

**Figure 9** Difference in MPs content between hot spots and non-hot spots per distance at Beach D.

The partial regression coefficients $b_0$ and $b_1$ were −33.3 and 0.138, respectively. The relationship between MPs content and $P$ is shown in Fig. 10. The MPs content $x$ at $P = 0.2$, 0.5, and 0.8 was 230, 240, and 250 mg-MPs/m$^2$-sand, respectively. When the MPs content in the sand exceeded 250 mg-MPs/m$^2$-sand, the area visually determined to be HS. Discriminatory accuracy = (number of samples classified correctly)/(total number of samples) × 100 = 71/72 ×100 = 99%. In other words, when sampling squares are classified as HS above around 200 mg-MPs/m$^2$-sand or nHS below around 200 mg-MPs/m$^2$-sand, they can be discriminated with high accuracy by visual determination. Since the criteria for visual nHS determination is ''no MPs found'', a sand with MPs content less than 200 mg-MPs/m$^2$-sand would be considered clean for landscape purposes.

## Problems of this study

The author believes that MPs investigation at a fixed spot is not effective in terms of MPs accumulation rate. The author found that only the fixed spot had more litter than its neighbors, or conversely, the neighborhood was dirty and the fixed spot was clean. This is also explained in Section ''nHS is immediately adjacent to HS''. This means that the MPs measurement results at fixed spots are not representative. Although a great deal of effort is required to obtain the representative value of MPs accumulation rate in the study site by HS investigation, a meaningful accumulation rate will not be obtained unless the accumulation rate is calculated by obtaining the representative value multiple times. In the fixed spot investigation, MPs are reduced to zero because the MPs are collected and

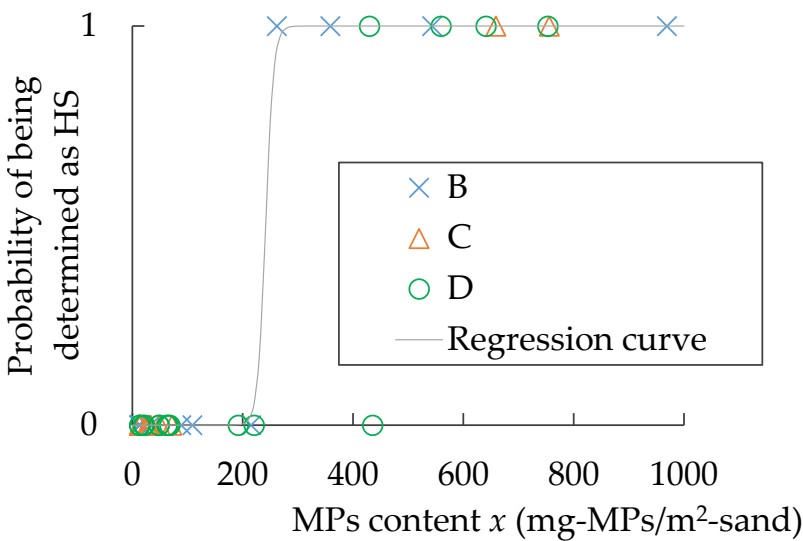

**Figure 10** Relationship between MPs content $x$ and probability of being determined as HS ($n = 72$) at Beaches B, C, and D. For $x > 1000$ ($n = 26$), all probability = 1.

brought back to the laboratory, but in the HS investigation, MPs in the entire study site are not reduced to zero because only a small portion of MPs samples are brought back.

## Future research

Research is needed to effectively evaluate the rate of accumulation. One of the solutions to reduce the overall MPs of a study site to zero is to conduct the HS investigation at the same time as a beach cleanup. Since public beaches are frequently subjected to beach cleanups, it is also useful to investigate MPs collected at marine litter deposition sites.

## Significance of this study

The weighting of the MPs content in HS and nHS by their respective areas enabled us to obtain the representative value and the dispersion of the MPs content in the entire study site as "average and error (mg-MPs/m²-sand)". This weighting method yields the true representative value that cannot be obtained by averaging multiple fixed spot investigations or multiple HS investigations on the high tide line.

As shown in Fig. 7, there is a large difference in MPs content between HS and nHS. HS and nHS are determined visually by students who have been engaged in MPs research for approximately one year. Furthermore, the students have not been specifically trained in visual determination. The author only instructed them to "visually determine where there is a large amount of MPs and where there is no MPs". Nevertheless, the MPs contents in HS and nHS showed a distinct difference, as expected. This means that the cases in which MPs content was high in places where the non-experts thought MPs was high by visual inspection and low in places where they thought MPs was low were repeated with a high degree of reproducibility. In other words, the accuracy of visual determination of MPs abundance on sandy beaches is high. The possibility that the MPs content can be

determined visually is also supported by Fig. 10. A portion of the high tide lines on the beach had gathered shells but not MPs, so that portion was actually nHS even though it was determined as nHS by the students. The MPs contents in HS and nHS can be determined visually.

## CONCLUSIONS

The following results were obtained for the proposed method to represent the results of investigation of MPs content in sand on sandy beaches in Nagasaki Prefecture, Japan.

1. The MPs accumulation rate in the study site was measured by periodic investigation at fixed spots. The average in the supratidal zone was $1.5 \pm 0.9$ mg-MPs/(m$^2$-sand·d) ($n = 15$).
2. The weighting of the MPs content in hot spots and non-hot spots by their respective areas enabled us to obtain the representative value and the dispersion of the MPs content in the entire study site as "average and error (mg-MPs/m$^2$-sand)". The MPs contents in the three beaches were $298 \pm 144$, $1,115 \pm 518$, and $4,084 \pm 2,243$ mg-MPs/m$^2$-sand, respectively. Using these values, it is possible to compare the MPs contents of multiple beaches.

## ACKNOWLEDGEMENTS

Special thanks are extended to Mr. Soichiro Yamamoto, Mr. Hayata Goto, Mr. Toshiki Sato, Ms. Yui Chinju, and Ms. Yukari Tokiyasu.

### Funding

This research was supported by a Grant-in-Aid for Scientific Research C (20K12208) from the Japan Society for the Promotion of Science (JSPS). The funders had no role in study design, data collection and analysis, decision to publish, or preparation of the manuscript.

### Grant Disclosures

The following grant information was disclosed by the author:
A Grant-in-Aid for Scientific Research C from the Japan Society for the Promotion of Science (JSPS): 20K12208.

### Competing Interests

The author declares that they have no competing interests.

### Author Contributions

- Hiroshi Asakura conceived and designed the experiments, performed the experiments, analyzed the data, prepared figures and/or tables, authored or reviewed drafts of the article, and approved the final draft.

## Field Study Permissions

The following information was supplied relating to field study approvals (*i.e.,* approving body and any reference numbers):

Field experiments were approved by Nagasaki Prefecture (project number: 2022-1).

## Data Availability

The raw measurements are available in the Supplementary File.

## Supplemental Information

Supplemental information for this article can be found online at http://dx.doi.org/10.7717/peerj.17207#supplemental-information.

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
