# Peer review of "Representation of investigation results of microplastics on sandy beaches—accumulation rate and abundance in the entire study site"

_PeerJ, doi:10.7717/peerj.17207_

## Round 0.1 · original submission · Major Revisions

The reviewers highlighted the importance of your submission but also identified a number of issues that make the manuscript unacceptable in its present form. The most important are (1) a revision of the abstract, (2) an update of your literature survey to consider also more recent studies, (3) a documentation of your data analysis in the Material and Methods section, (4) and improved presentation of the results and discussion. These and other points of reference are specifically detailed in the comments and suggestions of the three reviewers.

I hope that the comments of the reviewers will allow you to carry out a revision of the manuscript, which is a precondition for acceptance of the manuscript.

Reviewer 1 ·

Basic reporting

no comment

Experimental design

no comment

Validity of the findings

no comment

Additional comments

Representation of investigation results of microplastics on sandy beaches - accumulation rate and abundance in the entire study site has been presented. The subject is interesting and valuable. Therefore, the manuscript is potentially publishable. However, there are some points which should be addressed and discussed in the revised version, as mentioned below:

1. Please improve the abstract. Quantitative information should be provided in the abstract. The abstract should state briefly the purpose of the research, the principal results, and major conclusions. In the abstract, please add an indication of the achievements from your study that are relevant to the journal's scope. Please be concise - maximum 1-2 lines.
2. Please start the abstract by a short introduction of the current problem(s) and the solution, based on the current study, in one or two lines.
3. The introduction needs to be expanded to provide more background information on investigation of microplastics on sandy beaches. This will help contextualize the study and highlight its novelty.
4. The review of the literature needs more updating with works to have a clear and concise state-of-the-art analysis. This should more clearly show the knowledge gaps identified and link them to the paper's goals. The introduction section is poorly organized. While the general introduction is acceptable, the state-of-the-art review that follows is very difficult to understand and no specific thoughts can be inferred. The major defect of this study is the debate or argument is not clearly stated.
5. The authors should clearly explain the innovation and importance of their work on the introduction of the manuscript. They should justify the value of the work and compare their work with previously similar published papers. They should develop the advantage and applications of this procedure. The introduction section needs to be elaborated.
6. In the Introduction section, a last paragraph describing the main aim and novelty of this work is highly required. At the present form, the introduction section seems as an incomplete section without a suitable end.
7. Add a para about the source, size, abundance as well as distribution of microplastic in environment and health implications.
8. The authors should edit the manuscript more carefully as there are a few areas in the text where there are obvious mistakes or the meaning of the text can be easily misunderstood.
9. “other parameters were analyzed”. The other parameters which were analysed in the laboratory need to be mentioned here. Please enumerate them.
10. Data analysis. You conducted statistical analysis in your research, but you did not mention it in the Material and Method sections. Please tell me what statistical work you done and write me a section to be the “Data analysis” section.
11. Please provide the version number of the software you used and a URL for the software.
12. The authors should include the brand and model of all instruments they used in this project.
13. provide more details for these so-called instruments, like the dimensions, volume, components.
14. Please include the make, model and catalogue code.
15. Discussion part is not strong and every result may be supported with earlier/latest studies to prove and compare your findings.
16. RESULTS and DISCUSSION section. The authors gave discussion, but the discussion was not adequate and was not compared with similar discussion and results that were published by other authors. The authors used and cite some references, but it is not enough for suitable scientific discussion. The authors have to cite more related and relevant papers and give more discussion and evidence for theirs claims in the main text in the section.
17. The quality of the figures, graphs, plots, and images should be improved.
18. Please can the authors of the below paper confirm if they have received permission from all identifiable individuals to appear in Figure 4. Alternatively, this figure can be removed or replaced to not show the individuals.
19. Incorrect inter-relation between the data in the tables and the statements in the text.
20. All the findings of the current work need to be compared and discussed with the results of other researchers finding instead of having a general comparison with other researchers' works. The authors should perform a comparison between the forecasting results. In your discussion section, please link your empirical results with a broader and deeper literature review.
21. The authors should make comparison the method with similar methods that were described in literature. For the suggestion, the authors can make new table with discussion in the main text.
22. References format is not consistent. Add in more references of 2022 (better if you can have 2023 references)
23. The authors need to focus on the novelty of their own results, in particular, on the conclusion about the most effective concentration of alum and sand fraction for filtration. This emphasis must be duplicated in the "Abstract" and "Conclusions". You can start with words, for example, "Authors first identified ...", etc.
24. In Figure 8, the authors should give some statistical parameters such as p-value and the number of tests.
25. There are many old references. Some of the relevant following papers can be referred to and added to the revised manuscript:
1. Preliminary detection of microplastics in surface water of Maninjau Lake in Agam, Indonesia - ProQuest
2. Preliminary study on microplastic pollution in water and sediment at the Beaches of Pariaman City, West Sumatra, Indonesia - ProQuest
3. Short-term tourism alters abundance, size, and composition of microplastics on sandy beaches - ScienceDirect
4. Macro- and microplastic abundance from recreational beaches along the South Aegean Sea (Türkiye) - ScienceDirect
5. Presence of microplastics deposited in Sargassum sp. on sandy beaches - ScienceDirect

·

Basic reporting

The introduction section is really well written. However, the references should be more recent.

Experimental design

Very well written

Validity of the findings

Very well explained

Additional comments

The Abstract requires more information. I was wondering why the author did not specify the sampling location in the title or Abstract, as this would lead the reader to question where the sampling area is.
Overall, the manuscript is quite well written and provides significant information and explanations about the rate of microplastic accumulation on sandy beaches and worth for publication.

Reviewer 3 ·

Basic reporting

Kindly refer to the attached table for the detailed list of comments.

Experimental design

Kindly refer to the attached table for the detailed list of comments.

Validity of the findings

Kindly refer to the attached table for the detailed list of comments.

Additional comments

Kindly refer to the attached table for the detailed list of comments.

Annotated reviews are not available for download in order to protect the identity of reviewers who chose to remain anonymous.

---

## Round 0.2 · accepted · Accept

Thank you for the revision of the manuscript. I hereby certify that you have adequately taken into account the reviewers' comments, as I have checked by my own assessment of your revised manuscript. Based on my assessment as an Academic Editor, your manuscript is now ready for publication.

Reviewer 1 ·

Basic reporting

The reply to the queries is satisfactory.

Experimental design

The reply to the queries is satisfactory.

Validity of the findings

The reply to the queries is satisfactory.

Additional comments

-

·

Basic reporting

No comment

Experimental design

No comment

Validity of the findings

No comment

Additional comments

The author have made many corrections to improve the manuscript and it is worth of publication.